# Prognostic Value and Function of KLF5 in Papillary Thyroid Cancer

**DOI:** 10.3390/cancers13020185

**Published:** 2021-01-07

**Authors:** Poyil Pratheeshkumar, Abdul K. Siraj, Sasidharan Padmaja Divya, Sandeep Kumar Parvathareddy, Sarah Siraj, Roxanne Diaz, Rafia Begum, Saif S. Al-Sobhi, Fouad Al-Dayel, Khawla S. Al-Kuraya

**Affiliations:** 1Human Cancer Genomic Research, Research Center, King Faisal Specialist Hospital and Research Center, P.O. Box 3354, Riyadh 11211, Saudi Arabia; ppoyil@kfshrc.edu.sa (P.P.); asiraj@kfshrc.edu.sa (A.K.S.); pdivya@kfshrc.edu.sa (S.P.D.); psandeepkumar@kfshrc.edu.sa (S.K.P.); sarah-siraj@kfshrc.edu.sa (S.S.); rmelosantos87@kfshrc.edu.sa (R.D.); brafia@kfshrc.edu.sa (R.B.); 2Department of Surgery, King Faisal Specialist Hospital and Research Center, P.O. Box 3354, Riyadh 11211, Saudi Arabia; sobhi@kfshrc.edu.sa; 3Department of Pathology, King Faisal Specialist Hospital and Research Centre, P.O. Box 3354, Riyadh 11211, Saudi Arabia; dayelf@kfshrc.edu.sa

**Keywords:** papillary thyroid cancer, KLF5, HIF-1α, stemness, apoptosis, invasion

## Abstract

**Simple Summary:**

This study was conducted to investigate the clinical significance and prognostic value of KLF5 in a large cohort of Middle Eastern PTC patients and explore its functional role and mechanism in PTC cell lines in vitro and in vivo. We found KLF5 over-expression in PTC patient cases and this was significantly associated with aggressive clinico-pathological parameters and worse outcome. We also found a significant association between KLF5 and HIF-1α in PTC patients and cell lines. Functionally, KLF5 promoted cell growth, stemness, invasion, migration, and angiogenesis, while its inhibition reverses its action in PTC cell lines. Finally, the depletion of KLF5 regressed PTC tumor growth in nude mice. These data suggest that KLF5 may potentially be a suitable therapeutic target in PTC, and pharmacological inhibition of KLF5 might be a viable therapeutic option for the treatment of patients with an aggressive subtype of PTC.

**Abstract:**

The Krüppel-like factor 5 (KLF5), a zinc-finger transcriptional factor, is highly expressed in several solid tumors, but its role in PTC remains unclear. We investigated the expression of KLF5 protein in a large cohort of PTC patient samples and explored its functional role and mechanism in PTC cell lines in vitro and in vivo. KLF5 overexpression was observed in 65.1% of all PTC cases and it was significantly associated with aggressive clinico-pathological parameters and poor outcome. Given the significant association between KLF5 and HIF-1α overexpression in PTC patients, we investigated the functional correlation between KLF5 and HIF-1α in PTC cells. Indeed, the analysis revealed the co-immunoprecipitation of KLF5 with HIF-1α in PTC cells. We also identified KLF5-binding sites in the HIF-1α promoter that specifically bound to KLF5 protein. Mechanistically, KLF5 promoted PTC cell growth, invasion, migration, and angiogenesis, while KLF5 downregulation via specific inhibitor or siRNA reverses its action in vitro. Importantly, the silencing of KLF5 decreases the self-renewal ability of spheroids generated from PTC cells. In addition, the depletion of KLF5 reduces PTC xenograft growth in vivo. These findings suggest KLF5 can be a possible new molecular therapeutic target for a subset of PTC.

## 1. Introduction

Papillary thyroid cancer (PTC) is the most prevalent endocrine malignancy, accounting for approximately 80–90% of all thyroid cancer cases, with rapidly increasing incidence worldwide [1]. Although PTC is highly curable, 10–15% patients still have an aggressive course and become refractory to current therapeutic approaches [2,3]. Therefore, understanding the underlying molecular mechanisms that drive PTC aggressiveness can provide promising molecular and therapeutic targets that can help to improve patient survival, especially those with poor outcomes.

We have previously shown in PTC evolution, that an aggressive subset of patients exhibits aberrant expression of HIF-1α [4]. Interestingly, in our current large cohort of PTC, we find that this expression is strongly correlated with overexpression of Krüppel-like factor 5 (KLF5). KLF5 is a zinc-finger transcriptional factor that plays an important role in cell transformation, cancer stemness, angiogenesis and migration [5,6,7,8,9,10]. KLF5 expression and activity are altered in many human cancers [6,7,8,9,10,11,12,13]. KLF5 has been found to play an oncogenic role, where it is associated with tumor progression, aggressive clinical behavior and poor survival [6,7,11,12,13,14]. KLF5 has been identified as a trans-activator of HIF-1α [15,16]. A recent single study of 98 thyroid cancers (66 PTC, 27 follicular and 5 anaplastic) have explored the role of KLF5 in thyroid tumorigenesis [10]. However, a larger study to explore the clinical role, function, regulation and correlation with HIF-1α in PTC is needed.

Here, we have thoroughly investigated the clinical role of KLF5 in more than 1200 PTC and its correlation with other clinico-pathological parameters, including HIF-1α. Furthermore, we explored the function of KLF5 in PTC cell lines and its interaction with HIF-1α. By examining the impact of KLF5 in PTC cells, we provide strong evidence of the oncogenic role of KLF5 in PTC.

## 2. Results

### 2.1. KLF5 Expression in Normal Thyroid and PTC Tissues

KLF5 expression was determined in 1219 PTC and 225 normal thyroid tissues by immunohistochemistry using TMA. We found KLF5 to be significantly upregulated in PTC tissues, compared to normal thyroid tissues (*p* < 0.001; Figure 1A). The detailed association between KLF5 expression and the clinico-pathological parameters of patients with PTC is presented in Table 1. KLF5 overexpression was noted in 65.1% (793/1219) of PTC cases (Figure 1B), which was significantly associated with tall cell variant (*p* < 0.0001), extra-thyroidal extension (*p* = 0.0003), lymph node metastasis (*p* < 0.0001) and stage IVA tumors (*p* = 0.0003). The protein levels of KLF5 were also significantly correlated with HIF-1α expression (*p* = 0.0492). A significant association was also noted between KLF5 overexpression and *BRAF* mutation (*p* < 0.001). Interestingly, overexpression of KLF5 was significantly associated with disease-free survival in univariate analysis (*p* = 0.0066; Table 1, Figure 1C), although significance was not retained upon multivariate analysis, after adjusting for confounding factors such as age, gender, histology, extra-thyroidal extension, and stage of tumor.

### 2.2. KLF5 Functions Upstream of HIF-1α.

We showed a significant association of KLF5 overexpression and HIF-1α in our PTC patient cohort. To investigate the KLF5 and HIF-1α association in vitro, we examined the basal level of KLF5 and HIF-1α in PTC cell lines by immuno-blotting. We identified two PTC cell lines (BCPAP and TPC-1) expressing both KLF5 and HIF-1α, and one PTC cell line (K1) with low expression (Figure 2A). We performed co-immunoprecipitation (IP) analysis to determine the physical interaction of KLF5 with HIF-1α using anti-KLF5 and anti-HIF-1 α antibodies. The result shows that KLF5 binds to HIF-1α 1 (Figure 2B), and vice versa (Figure 2C), in BCPAP and TPC-1 cell lines. We also found that the pharmacological inhibition of KLF5 using ML264 markedly decreased the physical binding of KLF5 with HIF-1α in PTC cells (Figure 2D).

In an attempt to determine whether KLF5 transcriptionally activates HIF-1α in PTC cell lines, we performed chromatin immunoprecipitation (ChIP) assay. We identified four KLF5-binding sites (Figure 2E) in the 2-kb *STAT3* promoter. Our analysis showed that KLF5 binds to *HIF-1α* promoters at all the sites, S1 (−481–486), S2 (−602–607), S3 (−931–936) and S4 (−1552–1557) in PTC cells. In addition, ML264 treatment decreased the degree of KLF5 binding to HIF-1α promoters at all the sites in a dose course (Figure 2E,F).

We also inhibited KLF5 expression using a specific siRNA or KLF5 inhibitor, ML264, and examined the expression of KLF5 and HIF-1α in PTC cells. The inhibition of KLF5 using siRNA or ML264 noticeably down-regulated KLF5 and HIF-1α expressions in PTC cells (Figure 2G,H). We also knockdown HIF-1α to see the effect on KLF5 expression. Figure 2I shows that knockdown of HIF-1α markedly down-regulated the expression of HIF-1α, while KLF5 expression remained unchanged. To further verify the KLF5 and HIF-1α association in vitro, we overexpressed KLF5 in a low-expressing cell line (K1) and analyzed the HIF-1α expression. Figure 2J shows that forced expression of KLF5 dramatically increased the HIF-1α expression in K1 cell line. All these results indicate the in vitro association of KLF5 with HIF-1α, where KLF5 was found to be a functional upstream of HIF-1α.

### 2.3. Downregulation of KLF5 Inhibits Tumor Cell Invasion, Migration, and Angiogenesis

KLF5 has been associated with cancer cell invasion, migration and angiogenesis [7,8]. We sought to determine the effect of KLF5 downregulation on migration, invasion and angiogenesis in PTC cell lines. KLF5 inhibition using ML264 or siRNA significantly reduced invasion (Figure 3A–C) and migration (Figure 3D,E) of PTC cell lines. Moreover, ML264 treatment or KLF5 siRNA knockdown markedly down-regulated the expressions of KLF-5, HIF-1α, MMP-2, MMP-9 and VEGF in PTC cell lines (Figure 3F,G). The above results indicate that the silencing of KLF5 with siRNA exhibited similar results of inhibiting KLF5 with ML264. Conversely, forced expression of KLF5 in K1 cells (KLF5 low expressing cells) increased invasion (Appendix A) and migration (Appendix A), as well as the expressions of KLF5, HIF-1α, MMP-2, MMP-9 and VEGF in these cell lines (Appendix A). We also determined the effect of KLF5 inhibition on angiogenesis by analyzing the endothelial cell tube formation, which is one of the key steps of angiogenesis [17]. Human umbilical vein endothelial cells (HUVEC) were cultured on a thin layer of matrigel with supplemented with condition media (source of VEGF) from PTC cells with and without ML264 treatment. HUVECs formed elongated tube-like structures when cultured in condition media from untreated PTC cells, whereas condition media from ML264-treated PTC cells successfully reduced the endothelial cell tube formation (Figure 3H,I). Furthermore, we performed the ELISA experiment to determine the secreted levels of VEGF and MMP-9 in a cell culture supernatant with and without ML264 treatment in PTC cells. We found reduced levels of MMP-9 (Appendix A) and VEGF (Appendix A) secretion in ML264 treated PTC cells. These findings demonstrate that inhibition of KLF5 reduces PTC cell invasion, migration and angiogenesis by down-regulating MMP-2, MMP-9 and VEGF.

To verify the role of HIF-1α in KLF5-induced invasion and migration, we silenced HIF-1α using siRNA and analyzed the invasive and migratory potential of PTC cells. The silencing of HIF-1α significantly decreased the invasive (Appendix A) and migratory (Appendix A) potential of both tested PTC cells. Furthermore, the silencing of HIF-1α dramatically down-regulated the expressions of HIF-1α, MMP-2, MMP-9 and VEGF, while KLF5 expression remained unchanged (Appendix A). These data clearly indicate that KLF5-induced invasion and migration is mediated via HIF-1α.

### 2.4. Downregulation of KLF5 Inhibits PTC Cell Growth In Vitro

Overexpression of KLF5 has been associated with poor survival and tumor progression in various cancers [6,14]. Therefore, we investigated whether pharmacological inhibition of KLF5 using ML264 would be a possible therapeutic approach to attenuate cell growth and persuade apoptosis in PTC cells. We treated PTC cell lines with ML264 at different doses for 48 h to determine the cell viability. ML264 treatment significantly decreased the cell viability in both the cell lines, as shown by MTT assay (Figure 4A). In addition, ML264 treatment (Figure 4B,C) or KLF5 knockdown (Figure 4D) decreased the clonogenecity of PTC cells. To test whether ML264-induced growth inhibition was due to apoptosis, we treated PTC cells with ML264 at different doses for 48 h and analyzed the cells for apoptosis after labeling with Annexin V and PI l, followed by flow cytometry. There was a significant increase in the percentage apoptosis after treatment with ML264 in PTC cells (Figure 4E). Our clinico-pathological data showed a significant association between KLF5 overexpression and activated AKT. We therefore examined the activation status of AKT in PTC cells following treatment with ML264 (5 and 10 μM) for 48 h. As shown in Figure 4F, there was downregulation of KLF-5, HIF-1α and inactivation of AKT in ML264 treated PTC cells, as determined by Western blotting. Furthermore, the treatment of ML264 successfully down regulated the expression of anti-apoptotic proteins, Bcl-2 and Bcl-xl as well as cleavage of caspase-3 and PARP in PTC cell lines (Figure 4F).

### 2.5. Inhibition of KLF5 Decreases the Self-Renewal Ability of Spheroids Generated from PTC Cells

KLF5 overexpression has been associated with stemness and self-renewal of cancer stem cells [9]. To test the role of KLF5 in spheroid growth in PTC, we generated spheroids from PTC cells and stemness of the spheroids were confirmed using stem cell markers (Figure 5A,B). There was a dramatic increase in the expression of KLF-5, HIF-1α and stem cell markers, CD44, CD133, NANOG, and OCT4 as well as ALDH activity in spheroids compared to respective adherent cells (Figure 5A,B). Next, we silenced KLF5 in PTC cells and grown in spheroid medium. Interestingly, the silencing of KLF5 significantly decreased the spheroid growth (Figure 5C,D) and stemness properties (Figure 5E,F). Furthermore, forced expression of KLF5 in K1 cells (KLF5 low-expressing cells) showed increased spheroid growth (Appendix A) and upregulated expression of KLF5, HIF-1α, CD44, CD133, NANOG, and OCT4 compared to empty vector transfected cells (Appendix A). To validate the role of HIF-1α in KLF5-induced spheroid growth, we silenced HIF-1α using siRNA and analyzed the spheroid growth in PTC cells. As shown in Appendix A, knockdown of HIF-1α significantly decreased the spheroid growth of both tested PTC cells. The above results clearly indicate the role of KLF5 in stemness maintenance in PTC cells and are mediated via HIF-1α.

### 2.6. Downregulation of KLF5 Inhibits PTC Cell Growth In Vivo

We showed that inhibition of KLF5 prominently attenuated PTC cell growth in vitro. Therefore, we wanted to investigate the effect of pharmacological inhibition of KLF5 on PTC tumor growth in vivo. For this, TPC-1 cells (4 × 10^6^ cells) were subcutaneously injected into the flanks of 6-week-old nude mice. When the tumor reached 100 mm^3^ diameter, mice were treated intraperitoneally with ML264 (10 and 25 mg/kg body weight) for 30 days. DMSO (0.1%, i.p) was served as vehicle control. Tumor volume was measured every week and mean tumor volume in each group (*n* = 6) of mice was calculated. After four weeks treatment, mice were sacrificed and tumors were excised and weighed. We showed that ML264 treatment significantly reduced the tumor growth, as shown by the decrease in tumor volume (Figure 6A) and tumor weight (Figure 6B) without effecting the body weight of mice (data not shown). As shown in Figure 6C, ML264 treatment caused the shrinkage of tumor size. Proteins were isolated from excised tumors and analyzed the expressions of KLF5, HIF-1α, pAKT, MMP-2, MMP-9, VEGF, PARP and caspase-3 proteins. As shown in Figure 6D, there was downregulation of KLF5, HIF-1α, pAKT, MMP-2, MMP-9 and VEGF, with cleavage of PARP and caspase-3 following ML264 treatment. These data suggest that targeting KLF5 can attenuates tumor growth in vivo.

## 3. Discussion

In this study, we reveal that KLF5 is highly expressed in 65.1% of PTC tumor samples (793/1219). The upregulation of KLF5 is found to be significantly associated with several aggressive clinico-pathological parameters such as tall cell variant, the presence of extrathyroidal extension, higher stage and lymph node metastasis. A previous study has revealed, in a small number of PTC tumor samples, an association between KLF5 and lymphnode metastasis, but not other clinical parameters, which could be due to the small number of patient samples in their cohort [10]. The prognostic significance of KLF5 upregulation seen in our cohort is similar to previously reported studies in other organ sites, like pancreatic [6], gastric [18] and lung cancers [19]. However, KLF5 upregulation is not an independent prognostic factor in multivariant analysis.

To investigate whether the clinically significant association between KLF5 upregulation and BRAFV600E mutation (which is the most potent activator of MAPK pathway) has impacted PTC patients’ survival, an additional study is currently underway to obtain further insight about this intriguing finding. Our study provides several lines of evidence supporting the oncogenic role of KLF5 in PTC. First, KLF5 ablation drastically reduces PTC cell proliferation in vitro and PTC xenograft growth in vivo. Second, it also demonstrates that KLF5 downregulation inhibits tumor growth by increasing apoptosis, as revealed by the downregulation of anti-apoptotic factors, bcl-2 and bcl-xL as well as cleavage of PARP and caspase-3 in both PTC cell lines. Furthermore, the inhibition of KLF5 markedly decreased MMP-2, MMP-9 and VEGF, which further confirms the role of KLF5 not only in tumor growth, but also in migration and angiogenesis and overall metastatic potential of PTC cells. These findings are consistent with previous studies that showed the role of KLF5 and the importance of its inhibition in tumor growth from other organ sites [20,21,22,23]. More importantly, ML264 treatment attenuated PTC growth in vitro and in vivo, which supports the earlier findings in colorectal cancer [23] and osteosarcoma [22]. Therefore, we suggest that the pharmacological inhibition of KLF5 might be a promising therapeutic strategy for PTC patients exhibiting aggressive subtypes.

Another interesting finding of this study is the identification of a biological relationship between KLF5 and the transcription factor HIF-1α in PTC. Our data suggest that KLF5 modulates the expression of HIF-1α in PTC. First, functional studies show KLF5 co-immunoprecipitates with HIF-1α, which is consistent with previous studies in lung cancer cells [16]. Importantly, we showed that KLF5 binds to the HIF-1α promoter in PTC cells using ChIP analysis and this binding was disrupted with ML264. Second, knockdown of KLF5 decreases HIF-1α expression. Interestingly, HIF-1α inhibition has no effect on KLF5 expression, suggesting that KLF5 functions upstream of HIF-1α.

Our data also show a potential role of KLF5 in the regulation of cancer-stem-cell-like cells in PTC. Silencing of KLF5 significantly suppressed the spheroid growth and downregulated the expression of stem cell markers such as CD44, CD133, NANOG and OCT4 in PTC. This is in accordance with a previous report that revealed the role of KLF5 in maintaining stemness properties in hepatocellular carcinoma [9]. Enhanced KLF5 expression increases PTC cell growth and colony formation.

## 4. Materials and Methods

### 4.1. Sample Selection

Archival samples from 1219 PTC patients diagnosed between 1994 and 2016 at King Faisal Specialist Hospital and Research Center (Riyadh, Saudi Arabia) were included in the study. Clinicopathological data are summarized in Table 2. Tumors were staged according to the 8th edition of the AJCC staging system. Two hundred and twenty-five normal thyroid tissues were also included in the study. Extra-thyroidal extension included both microscopic and macroscopic extension beyond the thyroid. The Institutional Review Board of the hospital provided approval for the collection of archival samples. For this study, since only archival paraffin tissue blocks were used, a waiver of consent was obtained from Research Advisory Council (RAC) on 13 November 2017 (RAC#2170022).

### 4.2. Tissue Microarray Construction and Immunohistochemistry

All samples were analysed in a tissue microarray (TMA) format. TMA construction was performed as described earlier [24]. Briefly, tissue cylinders with a diameter of 0.6 mm were punched from representative normal and tumor regions of each donor tissue block and brought into a recipient paraffin block using a modified semiautomatic robotic precision instrument (Beecher Instruments, Woodland, WI, USA). Two cores of PTC were arrayed from each case.

Standard protocol was followed for imunohistochemistry staining. For antigen retrieval, Dako (Dako Denmark A/S, Glostrup, Denmark) Target Retrieval Solution pH 9.0 (Catalog number S2368) was used, and the slides were placed in Pascal pressure cooker at 120 °C for 10 min. The slides were incubated with primary antibody against KLF5 (ab137676, 1:800 dilution, pH 6.0, Abcam, Cambridge, UK) and HIF-1α (1:100 dilution, pH 6.0, Novus Biologicals, Littleton, CO, USA). The Dako Envision Plus System kit was used as the secondary detection system with 3, 30-diaminobenzidine as chromogen. All slides were counterstained with hematoxylin, dehydrated, cleared and mounted. Normal tissues of different organ systems were also included in the TMA to serve as positive controls. Negative control was performed by omission of the primary antibody. Only fresh cut slides were stained simultaneously to minimize the influence of slide aging and maximize the reproducibility of the experiment. The slides were independently examined by two pathologists. If there was a discrepancy in the individual scores, both pathologists carried out a re-evaluation until a consensus was reached.

KLF5 immunohistochemical expression was seen predominantly in the nuclear compartment, and nuclear expression was quantified using the proportion score as described previously [25]. Briefly, the proportion of positively stained tumor cells was calculated as a percentage for each core and the scores were averaged across two tissue cores from the same tumour to yield a single percent staining score representing each cancer patient. Cases showing an expression level of more than 10% were classified as overexpression, and those with ≤10% as low expression. For HIF-1α, median proportion score was used as described previously [26]. Briefly, the median expression level of HIF-1α in our cohort was 20%, and hence used as a cut-off for over-expression (>20%).

### 4.3. Cell Culture

The PTC cell line, BCPAP, was obtained from DSMZ, and TPC-1 was kindly provided by Dr. Bryan McIver (Department of Endocrinology, Mayo Clinic, Rochester, Minnesota). K1 cell line was purchased from American Type Culture Collection (ATCC). Cell lines were cultured in RPMI 1640 media supplemented with 10% fetal bovine serum (FBS),100 Units/mL penicillin/streptomycin and 100 Units/mL Glutamine as described previously [27]. These cell lines were authenticated in-house using short tandem repeats PCR and the results were in accordance with published data [17,28]. All experiments were performed using 5% FBS in RPMI 1640 media.

### 4.4. Reagents and Antibodies

ML264 (KLF5 selective inhibitor) was purchased from MyBioSource, Inc. (San Diego, CA, USA). KLF5 antibody (HPA040398) was obtained from Sigma (St. Louis, MO, USA). Antibodies against HIF-1α (sc-13515), pAKT (sc-7985), VEGF (sc-57496), caspase-3 (sc-56053) and GAPDH (sc-25778) were purchased from Santa Cruz Biotechnology, Inc. (Santa Cruz, CA, USA). Antibodies against AKT (9272), MMP-2 (13132), MMP-9 (2270), Bcl-2 (2876), Bcl-xl (2762), PARP (9542), cleaved caspase-3 (9664), CD44 (3570), CD133 (64326), NANOG (4903) and OCT4 (75463) were purchased from Cell Signaling Technology (Danvers, MA, USA).

### 4.5. MTT (3-(4,5-Dimethylthiazol-2-yl)-2,5-Diphenyltetrazolium Bromide) Assays

PTC cells were incubated at the concentration of 10^4^ cells per well in a 96-well format. Cells were then treated with various doses of ML264 for 48 h in a final volume of 0.2 mL. Cell viability was measured by MTT cell viability assay. Six wells for each dosage including vehicle control were analyzed for each experiment.

### 4.6. Gene Silencing Using siRNA

*KLF5* siRNA, and scrambled control siRNA were purchased from OriGene (Rockville, MD, USA). *HIF-1α* siRNA, and scrambled control siRNA were purchased from Santa Cruz Biotechnology, Inc. (Santa Cruz, CA, USA). Cells were transfected using Lipofectamine 2000 (Invitrogen, Carlsbad, CA, USA) for 6 h, following which the lipid and siRNA complex was removed and fresh growth medium was added. After 48 h of transfection, cells were used for immunoblotting.

### 4.7. Plasmid and Transfection

Plasmid DNA encoding human *KLF5* and shRNA-targeting human KLF5 were purchased from Origene (Rockville, MD, USA). The overexpression of KLF5 in PTC cells were performed using Lipofectamine™2000 (Invitrogen, Carlsbad, CA) according to the manufacturer’s protocol. Briefly, PTC cells were seeded in 6-well culture plate; when approximately 50% confluent, cells were transfected with 4 μg plasmid. After 48 h of transfection, overexpression of KLF5 and knockdown of KLF5 protein production were confirmed by immunoblotting.

### 4.8. Cell Invasion and Migration Assays

Cell invasion and migration assays were performed as described previously [17]. Briefly, cells after treatment with ML264 or siRNA knockdown for 48 h, cells were re-counted and equal number of cells were seeded into Trans-well inserts, either uncoated (for migration assay) or coated (for invasion assay) with growth factor-reduced matrigel for 24 h. After incubation, cells were stained with Diff-Quick stain set (Fisher Scientific, Pittsburg, PA, USA), and photographed under a fluorescent microscope.

### 4.9. Sphere Forming Assay

PTC cells (500/well) were plated on Corning 24-well ultra-low attachment plates (Sigma-Aldrich) grown in serum free DMEM-F12 (ATCC) supplemented with B27 (Thermo Fisher, Waltham, MA, USA ), 20 ng/mL epidermal growth factor (Sigma-Aldrich, St Louis, MO, USA), 0.4% bovine serum albumin (Sigma-Aldrich, St Louis, MO, USA) and 4 μg/mL insulin (Sigma-Aldrich, St Louis, MO, USA). Fresh medium was supplemented every 2 days. The spheroids were counted and photographed at day 14. For secondary spheroid formation, the primary spheroids were dissociated into single cells, and cultured on 24-well ultra-low attachment plates using spheroid culture medium for another 10 days.

### 4.10. The ALDH Activity

The ALDH activity was determined by the Aldefluor assay kit (Stem Cell Technologies, Inc., Cambridge, MA, USA) according to the manufacturer’s instructions. Cells were trypsinized and incubated with activated ALDEFLUOR reagent for 50 min at 37 °C. Control samples incubated with the inhibitor, DEAB, were used to ensure identification of ALDHhigh and ALDHlow subpopulations. All the stained cells were analysed by flow cytometer.

### 4.11. Chromatin Immunoprecipitation (ChIP) Assay

ChIP analysis was performed using a Pierce TMA garose ChIP Kit (Thermo Scientific, Rockford, IL, USA). Sheared chromatin was diluted and immunoprecipitated with 2 μg of anti-KLF5 or control IgG antibody. DNA protein complexes were eluted from protein A/G agarose beads using a spin column and were reverse cross-linked by incubating with NaCl at 65 °C. The intensity of KLF5 binding to the HIF1-α promoter was analyzed by Applied Biosystems^®^ 7500 Fast Real-Time PCR Detection System with SYBR Green PCR master mix using following primer sequences, KLF5 binding to the HIF1-α promoter sites, S1 (GGGTG, −481 to −486): (F) CTTTCCTCCGCCGCTAAAC and (R) GGGTTCCTCGAGATCCAATG, S2 (CACCC, −602 to −607): (F) GATGCATGTTTGGGACCAG and (R) CTCACGTGCTCGTCTGTGTT, S3 (GGGTG, −931 to −936): (F) AAACTCCGCCACAGAAAAAC and (R) CAAGCCCTTCCTTTGGTCTC, S4 (CACCC, −1552 to −1557):(F) AGGCTTCTCCAGCCTCACAC and (R) TATAGAAGCATCAAACTCTGACAAGA. General PCR amplification also performed in a Mastercycler^®^ thermal cycler (Eppendorf, Foster City, CA).

### 4.12. Animals and Xenografts Study

Six-week-old nude mice were obtained from Jackson Laboratories (Bar Harbor, ME, USA) and maintained in a pathogen-free animal facility at least 1 week before use. All animal studies were done in accordance with institutional guidelines and the study was approved by the animal use and care committee (ACUC) of King Faisal Specialist Hospital and Research Center on 13 November 2017 (RAC#2170022). The animals were divided into three groups (six mice/group). Tumor xenografts were generated by subcutaneously injection of TPC-1 cells (4 × 10^6^ cells per mouse/100 μL) suspended in serum-free medium with matrigel matrix (BD Biosciences, 1:1 ratio) into the flanks of nude mice. When tumors grew to about 100 mm^3^, mice were given an intraperitoneal injection of ML264 (10 and 25 mg/kg), twice a week for 30 days. Tumor volume and the body weight of each mouse was measured every week. The mice were sacrificed after four weeks of ML264 treatment, and individual tumors were harvested and weighed, then snap-frozen in liquid nitrogen for storage.

### 4.13. Statistical Analysis

Contingency table analysis and Chi square tests were used to study the relationship between clinicopathological variables and protein expression. Disease free survival curves were generated using the Kaplan–Meier method, with significance evaluated using the Mantel–Cox log-rank test. The limit of significance for all analyses was defined as *p* value of <0.05; two-sided tests were used in these calculations. The JMP11.0 (SAS Institute Inc., Cary, NC, USA) software package was used for data analyses.

For all functional studies, data are presented as means ± SD of triplicates in an independent experiment, which was repeated for at least two times with the same results. Student *t*-test (two-tailed) was performed for statistical significance, with a *p* < 0.05 used as the cut-off.

## 5. Conclusions

In summary, this study provides clinical evidence that KLF5 is widely upregulated in PTC tumor tissues and provides mechanistic evidence for tumorigenic role of KLF5, as well as its role in tumor progression, metastasis and stemness in PTC. Our data also uncover the KLF5/HIF-1α axis that regulates PTC cell growth, suggesting that aberrant KLF5 may be a potential therapeutic target in a subset of PTC.

## Figures and Tables

**Figure 1 cancers-13-00185-f001:**
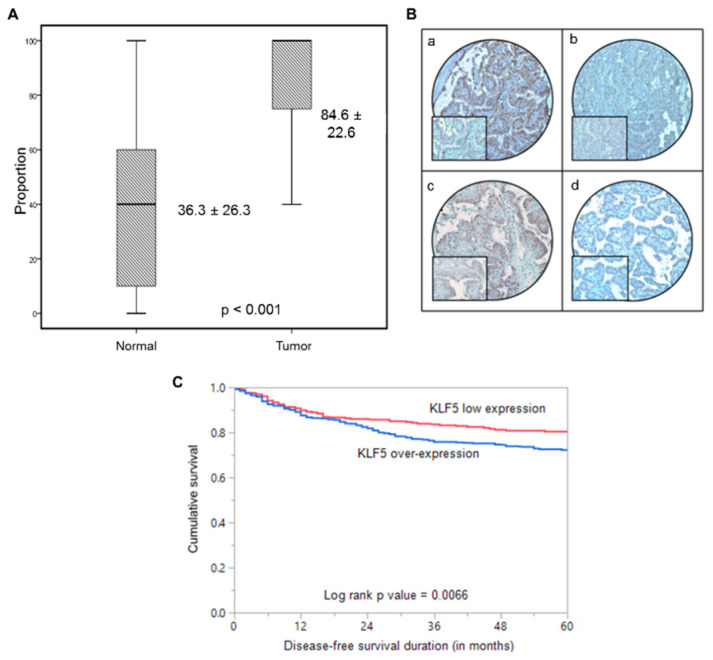
Immunohistochemical analysis of KLF5 and HIF-1α expression in Papillary Thyroid Cancer (PTC) TMA. (**A**) KLF5 expression in normal thyroid and papillary thyroid carcinoma (PTC) tissues. A significant difference in expression levels was noted between normal thyroid (*n* = 225) and PTC tissues (*n* = 1219) (*p* < 0.001). (**B**) PTC array spot showing overexpression of KLF5 (**a**) and HIF-1α (**c**). In contrast, another PTC tissue array spot showing low expression of KLF5 (**b**) and HIF-1α (**d**). 20X/0.70 objective on an Olympus BX 51 microscope. (Olympus America Inc, Center Valley, PA, USA) with the inset showing a 40X 0.85 aperture magnified view of the same TMA spot. (**C**) Kaplan–Meier survival analysis for the prognostic significance of KLF5 expression in PTC. PTC patients with overexpression of KLF5 had reduced disease-free survival at 5 years on univariate analysis compared to tumors showing low expression of KLF5 (*p* = 0.0066).

**Figure 2 cancers-13-00185-f002:**
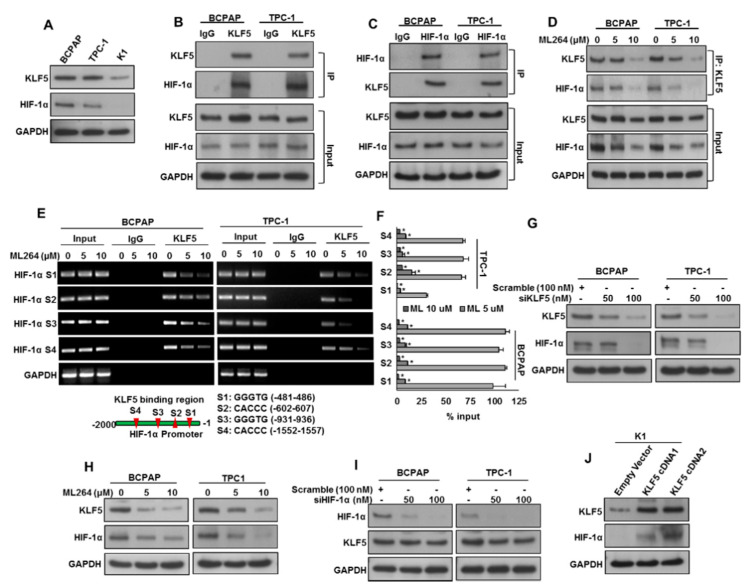
KLF5 is a functional upstream of HIF-1α. (**A**) Basal expression of KLF5 and HIF-1α in PTC cell lines. Proteins were isolated from three PTC cell lines and immunoblotted with antibodies against KLF5, HIF-1α and GAPDH. (**B**) KLF5 interact with HIF-1α. Cell lysates extracted from PTC cells were immunoprecipitated with KLF5 or IgG antibody. Interaction of endogenous KLF5 and HIF-1α was detected by immunoblotting. (**C**) HIF-1α interact with KLF5. Cell lysates extracted from PTC cells were immunoprecipitated with HIF-1α or IgG antibody. Interaction of endogenous KLF5 and HIF-1α was detected by immunoblotting. (**D**) ML264 disrupts the physical interaction of KLF5 and HIF-1α in PTC cells. PTC cells were treated with indicated doses of ML264 for 6 h. Cell lysates extracted from PTC cells were immunoprecipitated with KLF5 antibody. Interaction of endogenous KLF5 and HIF-1α was detected by immunoblotting. (**E**,**F**) KLF5 binding to *HIF-1α* promoter. For the ChIP assay, the KLF5-binding regions on *HIF-1α* promoter were identified. PTC cells were treated with and without ML264 (5 and 10 μm) for 6 h, fixed with formaldehyde, and cross-linked, and then chromatin was isolated. The chromatin was immunoprecipitated (IP) with an anti-KLF5 antibody or control mouse IgG. The KLF5 binding to the *HIF-1α* promoters was analyzed by regular PCR (**E**) or quantitative real-time PCR (**F**) with a primer specific to the KLF5-binding regions in *HIF-1α* promoter. The data represent the percent input and are normalized to each control. GAPDH was used as a loading control. (**G**) Silencing of KLF5 inhibits HIF-1α. PTC cells were transfected with scrambled siRNA and *KLF5* siRNA (50 and 100 nM). After 48 h, cells were lysed and proteins were immunoblotted with antibodies against KLF5, HIF-1α and GAPDH. (**H**) ML264 treatment down-regulates the expression of KLF5 and HIF-1α in PTC cells. PTC cells were treated with indicated doses of ML264 for 48 h. After cell lysis, equal amounts of proteins were separated by SDS-PAGE, transferred to immobilon membrane, and immuno-blotted with antibodies against KLF5, HIF-1α and GAPDH as indicated. (**I**) Knockdown of HIF-1α has no effect on KLF5 expression. PTC cells were transfected with scrambled siRNA and *HIF-1α* siRNA (50 and 100 nM). After 48 h, cells were lysedand proteins were immunoblotted with antibodies against HIF-1α, KLF5 and GAPDH. (**J**) Forced expression of KLF5 increases HIF-1α expression. K1 cells were transfected with either empty vector or *KLF5* cDNA for 48 h. Proteins were isolated and immunoblotted with antibodies against KLF5, HIF-1α and GAPDH for equal loading. Data presented in the bar graphs are the mean ± SD of triplicates in an independent experiments, which was repeated for at least two times with the same results. * Indicates a statistically significant difference compared to control with *p* < 0.05. Western blot experiments were repeated at least two times with the same results.

**Figure 3 cancers-13-00185-f003:**
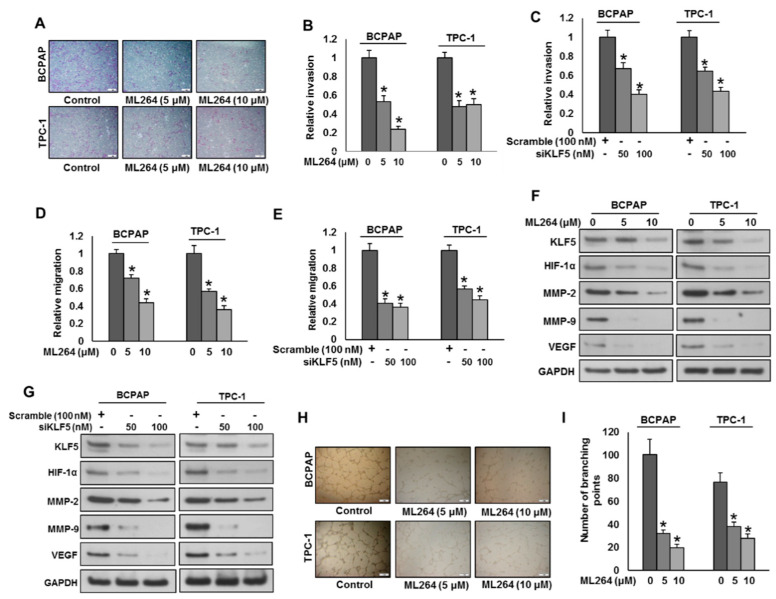
Downregulation of KLF5 inhibits tumor cell invasion, migration and angiogenesis. (**A**,**B**) KLF5 inhibition decreases the invasive capacity of PTC cells. PTC cells were treated with indicated doses ML264 and seeded into the upper compartment of invasion chambers. The bottom chambers were filled with RPMI media. After 24 h incubation, invaded cells were fixed, stained and quantified. (**C**) Silencing of KLF5 decreases invasion of PTC cells. PTC cells were transfected with scrambled siRNA and *KLF5* siRNA (50 and 100 nM). After 48 h, cells were seeded into the upper compartment of invasion chambers. The bottom chambers were filled with RPMI media. After 24 h incubation, invaded cells were fixed, stained and quantified. (**D**) KLF5 inhibition causes reduction in the migration capacity of PTC cells. PTC cells were treated with indicated doses ML264 and seeded into the upper compartment of migration chambers. The bottom chambers were filled with RPMI media. After 24 h incubation, migrated cells were fixed, stained and quantified. (**E**) Silencing of KLF5 decreases migration of PTC cells. PTC cells were transfected with scrambled siRNA and *KLF5* siRNA (50 and 100 nM). After 48 h, cells were seeded into the upper compartment of migration chambers. The bottom chambers were filled with RPMI media. After 24 h incubation, migrated cells were fixed, stained and quantified. (**F**) ML264 treatment down-regulates the expression of MMP-2, MMP-9 and VEGF in PTC cells. PTC cells were treated with indicated doses of ML264 for 48 h. After cell lysis, equal amounts of proteins were separated by SDS-PAGE, transferred to immobilon membrane, and immuno-blotted with antibodies against KLF5, HIF-1α, MMP-2, MMP-9, VEGF and GAPDH, as indicated. (**G**) Silencing of KLF5 down-regulates the expression of MMP-2, MMP-9 and VEGF in PTC cells. PTC cells were transfected with scrambled siRNA and *KLF5* siRNA (50 and 100 nM). After 48 h, cells were lysed and proteins were immunoblotted with antibodies against KLF5, HIF-1α, MMP-2, MMP-9, VEGF and GAPDH. (**H**,**I**) KLF5 inhibition decreases HUVECs tube formation. HUVECs grown on matrigel were treated with conditioned media from KLF5 treated and untreated PTCs for 24 h, cells were fixed, and tubular structures were photographed and quantified. Data presented in the bar graphs are the mean ± SD of triplicates in an independent experiment which was repeated for at least two times with the same results. * Indicates a statistically significant difference compared to control with *p* < 0.05. Western blot experiments were repeated at least two times with the same results.

**Figure 4 cancers-13-00185-f004:**
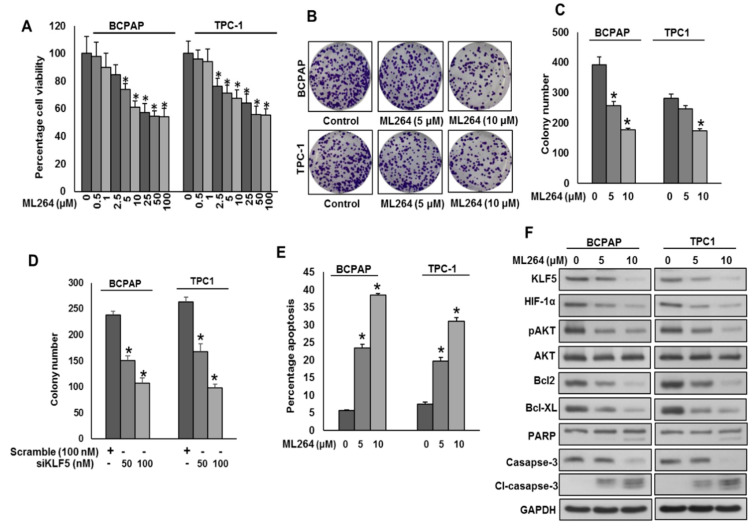
Downregulation of KLF5 inhibits PTC cell growth in vitro. (**A**) ML264 inhibits cell viability. PTC cells (10^4^) were incubated with indicated doses of ML264 for 48 h. Cell viability was performed using MTT. (**B**,**C**) ML264 inhibited clonogenicity. PTC cells (8 × 10^2^) after ML264 treatment were seeded into two dishes (60 mm diameter), and grown for an additional 10 days, then stained with crystal violet, and colonies were counted. (**D**) Knockdown of KLF5 decreases clonogenicity. PTC cells were transfected with scrambled siRNA and *KLF5* siRNA (50 and 100 nM). After 48 h, cells (8 × 10^2^) were seeded into two dishes (60 mm diameter), and grown for an additional 10 days, then stained with crystal violet, and colonies were counted. (**E**) ML264 induces apoptosis in PTC cell lines. PTC cells were treated with indicated doses of ML264 for 48 h and cells were stained with fluorescein-conjugated annexin-V and propidium iodide (PI) and analyzed by flow cytometry. (**F**) ML264 treatment causes inactivation of AKT and down-regulates the expression of anti-apoptotic proteins and induces the cleavage of caspase-3 and PARP. PTC cells were treated with indicated doses of ML264 for 48 h. After cell lysis, equal amounts of proteins were separated by SDS-PAGE, transferred to immobilon membrane, and immuno-blotted with antibodies against KLF5, HIF-1α, pAKT, AKT, Bcl-2, Bcl-xL, PARP, casapse-3, cleaved casapse-3 and GAPDH as indicated. All the experiments were repeated twice with similar results. Data presented in the bar graphs are the mean ± SD of triplicates in an independent experiment which was repeated at least two times with the same results. * Indicates a statistically significant difference compared to control with *p* < 0.05. Western blot experiments were repeated at least two times with the same results.

**Figure 5 cancers-13-00185-f005:**
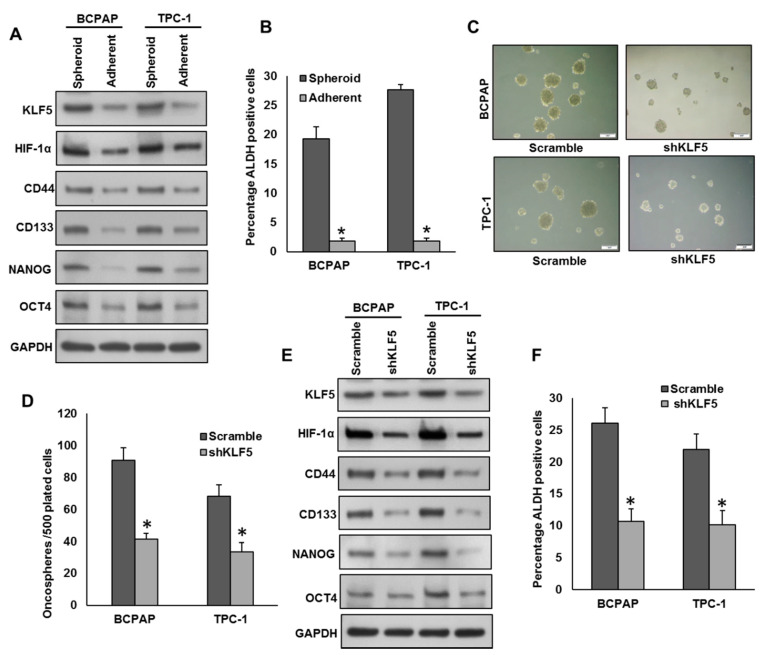
Inhibition of KLF5 decreases the self-renewal ability of spheroids generated from PTC cells. (**A**,**B**) Isolation of spheroid-forming cells from PTC cells. Sphere forming assay was performed by culturing PTC cells (5 × 10^2^ cells/well) in sphere medium for 14 days in 24-well ultra-low attachment plates. Proteins were isolated from spheroid-forming cells and respective parental adherent cells and immunoblotted with antibodies against KLF5, HIF-1α, CD44, CD133, NANOG, OCT4 and GAPDH (**A**). Spheroid-forming cells and adherent cells were labelled with Aldefluor with and without ALDH inhibitor, DEAB and analyzed by flow cytometer according to the manufacturer’s instructions (**B**). (**C**,**D**) Silencing of KLF5 inhibits self-renewal ability of spheroids. PTC cells were transfected with *KLF5* shRNA and cells were subjected to sphere forming assay. Spheroids in the entire well were counted. (**E**,**F**) Silencing of KLF5 inhibits stemness of spheroids as confirmed by immunoblotting using stem cell markers. PTC cells were transfected with scramble or *KLF5* shRNA’s and grown in sphere medium. Proteins were isolated from spheroids and immunoblotted with antibodies against KLF5, HIF-1α, CD44, CD133, NANOG, OCT4 and GAPDH (**E**). ALDH activity was also determined (**F**). Data presented in the bar graphs are the mean ± SD of triplicates in an independent experiment which was repeated at least two times with the same results. * Indicates a statistically significant difference compared to control with *p* < 0.05. Western blot experiments were repeated at least two times with the same results.

**Figure 6 cancers-13-00185-f006:**
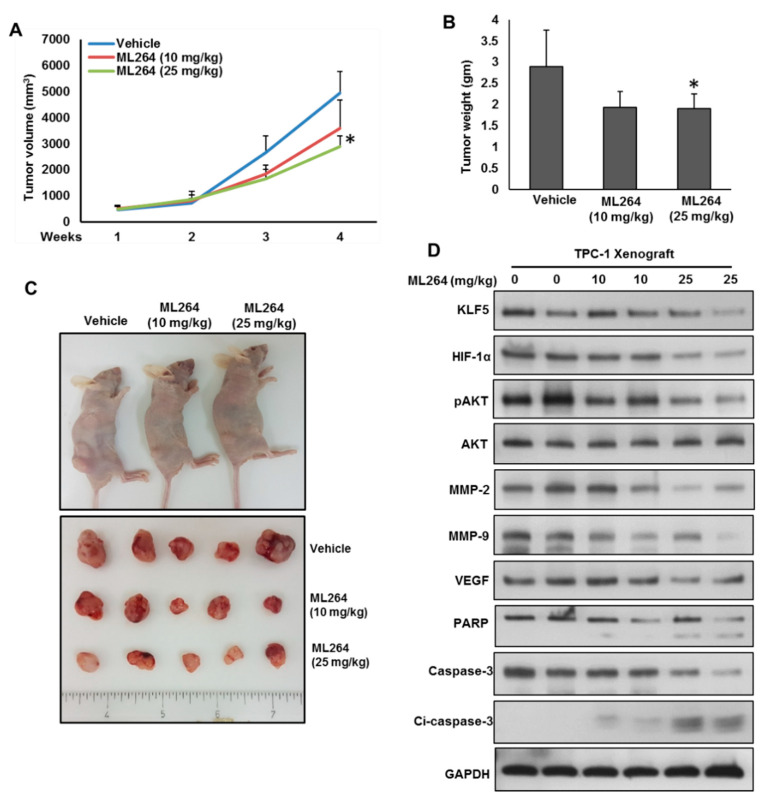
Downregulation of KLF5 inhibited PTC cell growth in vivo. TPC-1 cells were subcutaneously injected into the flanks of 6-week old nude mice (4 × 10^6^ cells per mouse). After tumors grew to about 100 mm^3^, mice were treated intraperitoneally with indicated dose of ML264, twice a week for 30 days. (**A**) The volume of each tumor was measured every week. The average (*n* = 6) tumor volume in each group of mice was calculated. (**B**) After four weeks treatment, mice were sacrificed and mean tumor weight (±SD) was calculated in each group, * *p* < 0.05. (**C**) Representative tumor images of each group of mice. (**D**) Tissue lysates from tumors were immuno-blotted with antibodies against KLF5, HIF-1α, pAKT, AKT, MMP-2, MMP-9, VEGF, PARP, caspase-3, cleaved caspase-3 and GAPDH.

**Table 1 cancers-13-00185-t001:** Clinico-pathological associations of KLF5 protein expression in PTC.

Clinico-Pathological Characteristics	Total		KLF5 Over-Expression		KLF5 Low Expression		*p*-Value
	No.	%	No.	%	No.	%	
**No. of patients**	1219		793	65.1	426	34.9	
**Age (Y)**							
<45	774	64.1	490	63.3	284	36.7	0.0858
≥45	434	35.9	296	68.2	138	31.8	
**Sex**							
Female	916	75.9	587	64.1	329	35.9	0.2151
Male	291	24.1	198	68.0	93	32.0	
**Histology Type**							
Classical Variant	839	73.9	554	66.0	285	34.0	<0.0001
Follicular Variant	191	16.8	102	53.4	89	46.6	
Tall-Cell Variant	105	9.3	83	79.0	22	21.0	
**Extrathyroidal extension**							
Absent	523	49.4	314	60.0	209	40.0	0.0003
Present	536	50.6	379	70.7	157	29.3	
**pT**							
pT1	307	26.1	186	60.6	118	39.4	0.1009
pT2	239	20.3	152	63.6	87	36.4	
pT3	482	41.0	330	68.5	152	31.5	
pT4	148	12.6	104	70.3	44	29.7	
**pN**							
pN0	553	46.8	313	56.6	240	43.4	<0.0001
pN1	628	53.2	453	72.1	175	27.9	
**pM**							
pM0	1149	95.5	747	65.0	402	35.0	0.8026
pM1	54	4.5	36	66.7	16	33.3	
**Stage**							
I	811	68.9	511	63.0	300	37.0	0.0003
II	94	8.0	54	57.4	40	42.6	
III	122	10.3	78	69.6	44	30.4	
IVA	114	9.7	95	83.3	19	16.7	
IVB	36	3.1	25	69.4	11	30.6	
**HIF-1α IHC**							
High	796	68.7	530	66.6	266	33.4	0.0492
Low	363	31.3	220	60.6	143	39.4	
**pAKT IHC**							
High	260	21.7	185	71.2	75	28.8	0.0279
Low	940	78.3	600	53.8	340	36.2	
**BRAF (V600E) mutation**							
Present	498	59.5	368	73.9	130	26.1	<0.0001
Absent	339	40.5	184	54.3	155	45.7	
**NRAS mutation**							
Present	43	5.1	20	46.5	23	53.5	0.0073
Absent	794	94.9	532	67.0	262	33.0	
**HRAS mutation**							
Present	24	2.9	16	66.7	8	33.3	0.9400
Absent	813	97.1	536	65.9	277	34.1	
**Disease Free Survival**							
5 years				72.3		80.4	0.0066

**Table 2 cancers-13-00185-t002:** Clinico-pathological variables for the patient cohort (*n* = 1219).

Clinicopathological Characteristics	*n* (%)
**Age**	
Median	38.0
Range (IQR) ^	29.0–51.0
**Gender**	
Female	925 (75.9)
Male	294 (24.1)
**Histopathology**	
Classical Variant	839 (68.8)
Follicular Variant	191 (15.7)
Tall Cell Variant	105 (8.6)
Other Variants	84 (6.9)
**Extra Thyroidal Extension**	
Absent	523 (42.9)
Present	536 (44.0)
Unknown	160 (13.1)
**pT**	
T1	307 (25.2)
T2	239 (19.6)
T3	482 (39.6)
T4	148 (12.1)
Unknown	43 (3.5)
**pN**	
N0	553 (45.4)
N1	628 (51.5)
Unknown	38 (3.1)
**pM**	
M0	1149 (94.3)
M1	54 (4.4)
Unknown	16 (1.3)
**Stage**	
I	811 (66.5)
II	94 (7.8)
III	122 (10.0)
IVA	114 (9.4)
IVB	36 (2.9)
Unknown	42 (3.4)

Abbreviations-^ Inter quartile range.

## Data Availability

The data presented in this study are available on request from the corresponding author.

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
