# Peer review of "Prognostic Value and Function of KLF5 in Papillary Thyroid Cancer"

_cancers, 2021, doi:10.3390/cancers13020185_

Round 1

Reviewer 1 Report

The authors have investigated the role of Krüppel-like factor 5 (KLF5)-a zinc finger transcription factor in thyroid cancer and the utility of this molecule as a therapeutic target. The role of KLF5 and its role as a therapeutic target has been established in other cancers predominately colorectal cancer and the authors have applied the same techniques described in other papers with respect to thyroid cancer. The manuscript follow a standard pattern of investigation establishing KLF5 expression in a large cohort of PTC by IHC and then knockdown of KLF5 invitro and invivo to establish it as a potential therapeutic target in PTC. A pathway through HIF-1αhas been demonstrated.
Comments.
1.The authors have asserted that KLF5 expression is associated with more aggressive cancer. I would like to see the correlation in other subtypes of PTC other than tall cell variant especially diffuse sclerosing variant or poorly differentiated PTC. The author further state that KLF5 overexpression correlates with extra-thyroidal extension. I would like them to explain is this microscopic ETE only or true macroscopic invasion of muscle , trachea or larynx.
2.Fig 1-“PTC patients with 86 overexpression of KLF5 had reduced disease-free survival at 5 years compared to tumors showing 87 low expression of KLF5 (p=0.0066). This needs to be altered as it did not hold up in multivariate analysis.
3.The authors need to further discuss the potential for ML264 as a therapeutic in humans. Intraperitoneal injection in a nude mouse with a small tumor volume may show an effect in vivo however this does not seem like a valid treatment for patients with widespread lung or bone metastatses refractory to radioiodine.

Author Response

Reviewer 1

The authors have investigated the role of Krüppel-like factor 5 (KLF5)-a zinc finger transcription factor in thyroid cancer and the utility of this molecule as a therapeutic target. The role of KLF5 and its role as a therapeutic target has been established in other cancers predominately colorectal cancer and the authors have applied the same techniques described in other papers with respect to thyroid cancer. The manuscript follow a standard pattern of investigation establishing KLF5 expression in a large cohort of PTC by IHC and then knockdown of KLF5 invitro and invivo to establish it as a potential therapeutic target in PTC. A pathway through HIF-1αhas been demonstrated.

Comments.

  1. The authors have asserted that KLF5 expression is associated with more aggressive cancer. I would like to see the correlation in other subtypes of PTC other than tall cell variant especially diffuse sclerosing variant or poorly differentiated PTC. The author further state that KLF5 overexpression correlates with extra-thyroidal extension. I would like them to explain is this microscopic ETE only or true macroscopic invasion of muscle, trachea or larynx.

We thank the reviewer for their suggestion. We agree with the reviewer that correlation of KLF5 expression in diffuse sclerosing variant and poorly differentiated thyroid cancer could provide further evidence for the association of KLF5 with more aggressive cancer. However, in our cohort, diffuse sclerosing variant comprised of only 8 cases and poorly differentiated thyroid cancer cases were not a part of this study since we focused only on the variants of PTC. Respecting the reviewer’s suggestion, we performed the correlation analysis for KLF5 in diffuse sclerosing variant of PTC. KLF5 over-expression was noted in 62.5% (5/8) cases and no associations were noted with any of the clinico-pathological parameters, probably due to the small sample size.

In our study, extra-thyroidal extension denotes either microscopic or macroscopic extension. Respecting reviewers comment, we have now incorporated this in the Methods section (Page 11, Line 342 - 343) as follows: “Extra-thyroidal extension included both microscopic and macroscopic extension beyond the thyroid”.

  1. Fig 1-“PTC patients with overexpression of KLF5 had reduced disease-free survival at 5 years compared to tumors showing low expression of KLF5 (p=0.0066). This needs to be altered as it did not hold up in multivariate analysis.

We thank the reviewer for their suggestion. We have now modified the legend for figure 1 as follows: “PTC patients with overexpression of KLF5 had reduced disease-free survival at 5 years on univariate analysis compared to tumors showing low expression of KLF5 (p=0.0066)”.

  1. The authors need to further discuss the potential for ML264 as a therapeutic in humans. Intraperitoneal injection in a nude mouse with a small tumor volume may show an effect in vivo however this does not seem like a valid treatment for patients with widespread lung or bone metastatses refractory to radioiodine.

We thank the reviewer for their suggestion to further discuss the potential for ML264 as a therapeutic option in humans. ML264 is a small molecule compound that inhibits the expression of KLF5 and its downstream targets [1]. In this study, we showed that ML264 treatment attenuates PTC growth in vitro and in vivo. In concordance with our findings, ML264 shown to suppress cellular proliferation and growth in many human cancers like colorectal cancer [2] and osteosarcoma [1] cells by inducing apoptosis and cell cycle arrest. Therefore, pharmacological inhibition of KLF5 might be a promising therapeutic strategy for PTC patients exhibiting aggressive phenotypes. We have now added this information in the Discussion section (Page no. 11, line 316-321). We agree with the reviewer that the results of in vivo effect of ML264 may not be valid for patients with widespread metastasis refractory to radioiodine. Further studies may be required to address this aspect.

References

  1. Huang, H.; Han, Y.; Chen, Z.; Pan, X.; Yuan, P.; Zhao, X.; Zhu, H.; Wang, J.; Sun, X.; Shi, P. ML264 inhibits osteosarcoma growth and metastasis via inhibition of JAK2/STAT3 and WNT/βcatenin signalling pathways. J. Cell. Mol. Med. 2020.
  2. de Sabando, A.R.; Wang, C.; He, Y.; García-Barros, M.; Kim, J.; Shroyer, K.R.; Bannister, T.D.; Yang, V.W.; Bialkowska, A.B. ML264, a novel small-molecule compound that potently inhibits growth of colorectal cancer. Mol. Cancer Ther. 2016, 15, 72-83.

Reviewer 2 Report

This manuscript states that PTC shows high expression of KLF5 and patients with KLF5 have a poor prognosis. In addition, cell line and animal experiments showed that KLF5 was involved in cell proliferation, invasion, migration, and angiogenesis, and that silencing of KLF5 showed tumor regression. It is very interesting paper, but consider the following:

  1. Figure 1c; I think the 5-year survival rate for papillary cancer is over 95%, but this graph shows that the survival rate is too low. Is there any reason?
  2. Table 1 Histological type; The percentage of tall cell variants is too high.
  3. Chemotherapy for thyroid carcinoma is for poorly differentiated and undifferentiated cancers. How about the expression of these tumors and KLF5?

Author Response

Reviewer 2

This manuscript states that PTC shows high expression of KLF5 and patients with KLF5 have a poor prognosis. In addition, cell line and animal experiments showed that KLF5 was involved in cell proliferation, invasion, migration, and angiogenesis, and that silencing of KLF5 showed tumor regression. It is very interesting paper, but consider the following:

We thank the reviewer for taking the time and effort to review our manuscript and for providing valuable suggestions to further improve the manuscript. It gives us great pleasure to know that the reviewer finds our study to be interesting. We have addressed the concerns of the reviewer point by point below.

  1. Figure 1c; I think the 5-year survival rate for papillary cancer is over 95%, but this graph shows that the survival rate is too low. Is there any reason?

We acknowledge the reviewer’s concern regarding survival rates in papillary thyroid cancer and agree with the reviewer that the 5-year overall survival rate in papillary thyroid cancer is more than 95%. However, we have evaluated the disease-free survival (DFS) rate in this study, which is defined as the length of time after primary treatment for a cancer ends that the patient survives without any signs or symptoms of that cancer. Similar to our findings, previous studies have also shown a lower DFS [1-4].

  1. Table 1 Histological type; The percentage of tall cell variants is too high.

We appreciate the reviewer’s concern regarding the high percentage of tall cell variants in our study. However, we would like to assure the reviewer that the proportion of tall cell variants in our study (9.1%) is in concordance with previously reported incidence for tall cell variant of PTC, which ranges from 7 – 14% [5-7].

  1. Chemotherapy for thyroid carcinoma is for poorly differentiated and undifferentiated cancers. How about the expression of these tumors and KLF5?

We thank the reviewer for their query. We indeed agree with the reviewer that chemotherapy in thyroid carcinoma is reserved for patients with poorly differentiated and undifferentiated cancers. However, our study included only papillary thyroid cancer and not the poorly differentiated or undifferentiated thyroid cancers. Hence, we did not analyze the expression of KLF5 in these subtypes of thyroid cancer.

References

  1. Henke, L.E.; Pfeifer, J.D.; Baranski, T.J.; DeWees, T.; Grigsby, P.W. Long-term outcomes of follicular variant vs classic papillary thyroid carcinoma. Endocrine connections 2018, 7, 1226-1235.
  2. Kim, S.Y.; Kim, B.-W.; Pyo, J.Y.; Hong, S.W.; Chang, H.-S.; Park, C.S. Macrometastasis in papillary thyroid cancer patients is associated with higher recurrence in lateral neck nodes. World Journal of Surgery 2018, 42, 123-129.
  3. Haugen, B.R.; Alexander, E.K.; Bible, K.C.; Doherty, G.M.; Mandel, S.J.; Nikiforov, Y.E.; Pacini, F.; Randolph, G.W.; Sawka, A.M.; Schlumberger, M. 2015 American Thyroid Association management guidelines for adult patients with thyroid nodules and differentiated thyroid cancer: the American Thyroid Association guidelines task force on thyroid nodules and differentiated thyroid cancer. Thyroid 2016, 26, 1-133.
  4. Jonklaas, J.; Sarlis, N.J.; Litofsky, D.; Ain, K.B.; Bigos, S.T.; Brierley, J.D.; Cooper, D.S.; Haugen, B.R.; Ladenson, P.W.; Magner, J. Outcomes of patients with differentiated thyroid carcinoma following initial therapy. Thyroid 2006, 16, 1229-1242.
  5. Boutzios, G.; Vasileiadis, I.; Zapanti, E.; Charitoudis, G.; Karakostas, E.; Ieromonachou, P.; Karatzas, T. Higher incidence of tall cell variant of papillary thyroid carcinoma in Graves' disease. Thyroid 2014, 24, 347-354.
  6. Axelsson, T.A.; Hrafnkelsson, J.; Olafsdottir, E.J.; Jonasson, J.G. Tall cell variant of papillary thyroid carcinoma: a population-based study in Iceland. Thyroid 2015, 25, 216-220.
  7. Song, E.; Jeon, M.J.; Oh, H.-S.; Han, M.; Lee, Y.-M.; Kim, T.Y.; Chung, K.-W.; Kim, W.B.; Shong, Y.K.; Song, D.E. Do aggressive variants of papillary thyroid carcinoma have worse clinical outcome than classic papillary thyroid carcinoma? European journal of endocrinology 2018, 179, 135-142.